# INFORMATION BOTTLENECK-INSPIRED EFFICIENT AND EXPLAINABLE FEDERATED ACTIVE LEARNING

## ABSTRACT

Federated learning (FL) enables collaborative model training on decentralized data while preserving privacy. Recently, explainable FL (XFL) has gained traction, aiming to generate semantically-rich latent representations that enhance interpretability of predictions. However, obtaining such representations typically requires large amounts of labeled data, which limits its applicability. Active learning, which reduces labeling cost by querying the most informative samples, is a promising solution. Existing federated active learning (FAL) methods mainly exploit model uncertainty for data selection. They mostly overlook the interactions and training dynamics of local and global models in data selection. This shortcoming can lead to suboptimal performance and reduced explainability in XFL settings. In this paper, we propose a novel explainable FAL framework - Federated Minimax Active Data Selection (`Fed-MADS`). The method leverages the information bottleneck technique to analyze model training dynamics, wherein a variational distribution is introduced and proposed to be implemented using the global model, making the approach well suited to the XFL setting. Then, a minimax objective is designed to identify unlabeled data points exhibiting significant divergence between local and global models in both latent representations and predicted labels. Extensive experiments on four benchmark datasets demonstrate that our method significantly outperforms state-of-the-art FAL approaches, achieving superior performance with fewer labeled data points.

## 1 INTRODUCTION

Federated Learning (FL) is a distributed collaborative machine learning paradigm that aims to train models across multiple data sources while maintaining data privacy (McMahan et al., 2017; Yang et al., 2019; Lim et al., 2020). To improve interpretability of FL, researchers have recently explored the applications of explainable AI methods (Koh et al., 2020; Barbiero et al., 2022) in FL setting to learn semantically-rich latent representations (Zhang & Yu, 2024). However, existing methods face two key challenges. First, learning such representations often requires large amounts of labeled data, as models must capture detailed relationships among inputs, outputs, and latent features. Second, the explicit introduction of latent representation learning alters the training dynamics of both local and global models, an aspect that has not been thoroughly studied. These issues limit the effectiveness of developing explainable FL (XFL) applications.

To address the first issue, Active Learning (AL) (Settles, 2009) has emerged as a promising strategy to reduce labeling costs by selectively querying labels for the most informative unlabeled data points. It has been well studied in various machine learning tasks. In recent years, many studies have attempted to integrate AL with FL, known as Federated Active Learning (FAL) (Wu et al., 2022; Kim et al., 2023; Zhang et al., 2023a; Cao et al., 2023), aiming to train effective FL models with fewer labeled data. However, existing FAL methods mainly exploit model uncertainty for data selection, which neglects the training dynamics of local and global models, as well as the interaction between data selection strategies and model interpretability. Consequently, directly applying existing methods to the XFL setting may lead to suboptimal performance and reduced explainability.

To address the second issue, Information Bottleneck (IB) principle(Tishby et al., 1999) provides a theoretical framework to explain the training dynamics for learning concise yet informative data representations (Tishby & Zaslavsky, 2015; Alemi et al., 2017). It posits that the optimal data

representation should maximize the mutual information between the latent representation and the output, while minimizing the mutual information between the input and the latent representation. The IB objective can be formulated as:

$$\min I(X, Z) - \beta I(Z, Y), \tag{1}$$

where $I(\cdot)$ denotes mutual information. $X$, $Y$ and $Z$ represent the domains of input, output and latent representation, respectively. $\beta$ is a trade-off hyperparameter. To the best of our knowledge, the IB principle has not been explored in XFL setting.

To address the above issues, this paper proposes an IB-inspired approach, namely Federated Minimax Active Data Selection (`Fed-MADS`) method, for explainable FAL. It leverages the IB principle to explain the training dynamics of explainable local and global models. Based on it, a minimax data selection objective is derived from the IB principle to efficiently select the most informative unlabeled data points from each FL client. A key innovation of `Fed-MADS` is that it implements the introduced variational distributions by both local and global parametric models, resulting in a seamlessly integration into XFL frameworks. Furthermore, it is designed to offer an intuitive interpretation: *it tends to select samples exhibiting large divergence in latent representations and final predictions between local and global models*. We conduct extensive experiments on four benchmark datasets commonly used in XFL to evaluate `Fed-MADS`. The results demonstrate that it significantly outperforms state-of-the-art FAL methods, incurring lower labeling costs while achieving competitive model performance.

## 2 RELATED WORKS

FL enables multiple decentralized clients to collaboratively train a global model while preserving data privacy by keeping data locally stored, addressing key privacy and security challenges (Konečný et al., 2016; McMahan et al., 2017; Yin et al., 2021). Recent works have introduced XFL (Zhang & Yu, 2024), by integrating explainable AI techniques into federated settings to enhance transparency and trustworthiness. Specifically, LR-XFL (Zhang & Yu, 2024) aims to learn a semantically-rich representation to form the logical explanations of the prediction. AL, on the other hand, enhances learning efficiency by selectively querying labels for the most informative unlabeled data points, significantly reducing labeling cost and accelerating convergence (Settles, 2009). To address the challenge of high labeling cost in FL, FAL (Wu et al., 2022; Kim et al., 2023) has recently emerged as an innovative research direction, integrating the decentralized data-handling paradigm of FL with the data-efficient querying strategies characteristic of AL.

Despite significant progress, existing FAL frameworks still face several critical challenges. First, most current approaches rely heavily on simplistic uncertainty-based querying strategies (Kim et al., 2022; Zhang et al., 2023b), which may not fully capture the complex data distributions inherent in federated setups. Second, practical deployment considerations, such as communication efficiency, labeling budget constraints, and robustness to client dropout, remain underexplored in current literature (Wu et al., 2022). To address these limitations, recent efforts have begun investigating more sophisticated sampling and adaptive querying algorithms tailored for the federated framework (Wang et al., 2019). However, existing approaches neglect model training dynamics and the interaction between data selection strategies and model interpretability, which may not fully utilize the information of XFL model, resulting in sub-optimal performance.

IB (Tishby et al., 1999) has been widely applied to explain the effectiveness of deep learning (Tishby & Zaslavsky, 2015; Alemi et al., 2017; Bang et al., 2021; Li et al., 2025) and federated learning (Uddin et al., 2022; Yang et al., 2023; Yan et al., 2025) in recent years. Some studies (Alemi et al., 2017; Bang et al., 2021) try to derive tractable variational approximations as objective functions for representation learning, enabling practical training of neural networks under information-theoretic constraints. Inspired by the same principle, concept bottleneck model (Koh et al., 2020) is proposed as an interpretable alternative, where models are trained to first predict human-understandable concepts before making final predictions. In addition to empirical studies, theoretical analyses have been conducted to better understand the implications and limitations of IB in deep learning (Kawaguchi et al., 2023). However, the application of IB theory to FAL remains underexplored.

# 3 THE PROPOSED Fed-MADS METHOD

## 3.1 PRELIMINARIES

We denote vectors by bold lowercase letters. Let $\mathcal{P}_X(\boldsymbol{x})$ be the distribution of random variable $X$. We assume that data are generated from a joint distribution $\mathcal{P}_{(X,Z,Y)}(\boldsymbol{x}, \boldsymbol{z}, y)$, where $\boldsymbol{x}$ is the input, $y$ is the output, and $\boldsymbol{z}$ is the latent representation. For brevity and clarity, we use $P(\boldsymbol{x}, \boldsymbol{z}, y)$ to denote this joint distribution when the context is evident. We follow the IB literature (Alemi et al., 2017) to define the joint distribution as $\mathcal{P}(\boldsymbol{x}, \boldsymbol{z}, y) = \mathcal{P}(\boldsymbol{x})\mathcal{P}(\boldsymbol{z}|\boldsymbol{x})\mathcal{P}(y|\boldsymbol{z})$ using Markov assumption, where $\mathcal{P}(\boldsymbol{z}|\boldsymbol{x})$ is the conditional distribution of $\boldsymbol{z}$ given $\boldsymbol{x}$.

Our study focuses on horizontal FL scenarios with i.i.d. data. AL is performed under the pool-based setting. Specifically, we assume that there are $K$ clients, each possessing a local dataset $\mathbb{D}_i = \{\mathbb{L}_i, \mathbb{U}_i\}$, where $\mathbb{L}_i = \{(\boldsymbol{x}_i^j, y_i^j)\}_{j=1}^{n_i^L}$ is a small initial labeled set and $\mathbb{U}_i = \{\boldsymbol{x}_i^j\}_{j=1}^{n_i^U}$ is a large unlabeled set, i.e., $n_i^L \ll n_i^U$. $\boldsymbol{x}_i^j$ is the input data, $y_i^j$ is the label, and $n_i^L$ and $n_i^U$ are the number of labeled and unlabeled data points in client $i$, respectively. The total number of data points in client $i$ is $n_i = n_i^L + n_i^U$.

## 3.2 FRAMEWORK AND PROCESS OF THE PROPOSED Fed-MADS

Without loss of generality, we consider the explainable model as a combination of an encoder and a decoder. In the proposed Fed-MADS framework, as shown in Figure 1, client $i$ first receives the global model $f_g = \{\boldsymbol{\mu}_1, \boldsymbol{\mu}_2\}$ from the server, where the global model consists of a parametric encoder $q^e(\boldsymbol{z}|\boldsymbol{x}; \boldsymbol{\mu}_1)$ and a decoder $q^d(y|\boldsymbol{z}; \boldsymbol{\mu}_2)$. Next, client $i$ trains a local model on the labeled dataset with the help of global model. The local model is parameterized by $f_i = \{\boldsymbol{\theta}_1, \boldsymbol{\theta}_2\}$, which consists of a parametric encoder $p_i^e(\boldsymbol{z}|\boldsymbol{x}; \boldsymbol{\theta}_1)$ and a decoder $p_i^d(y|\boldsymbol{z}; \boldsymbol{\theta}_2)$. Here,

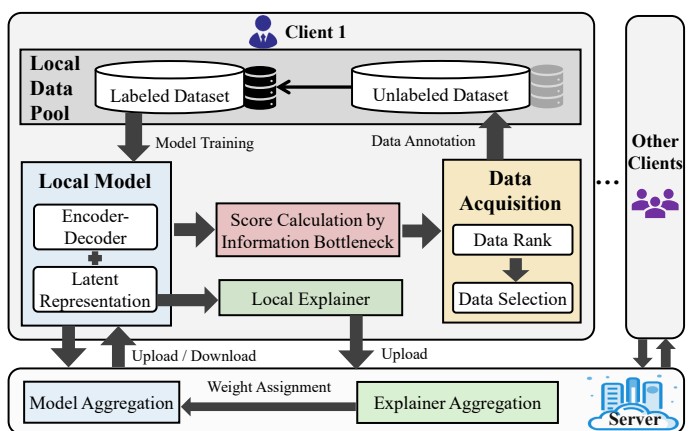

Figure 1: The overall framework of the proposed Fed-MADS method.

the encoder also serves as a local explainer to produce semantic information for prediction understanding. Subsequently, client calculates the utility score of each unlabeled data points $\forall \boldsymbol{x} \in \mathbb{U}_i$ using both local and global models, and selects a batch of informative data points $\mathbb{Q}_i$ from $\mathbb{U}_i$ for label querying. Based on the calculated final selection score, data annotation starts up on the local data pool. The labeled set $\mathbb{L}_i$ is updated to $\mathbb{L}_i \cup \mathbb{Q}_i$. After these local processes, the client uploads the local model parameters to the server for aggregation, including explainer aggregation and model aggregation. The whole procedures of Fed-MADS are repeated until model convergence or the labeling budget is exhausted.

## 3.3 DETAILED DESIGN

To design the query strategy of Fed-MADS, we first introduce the learning objective in FL. We employ IB principle to explain the learning dynamics in XFL. The IB objective function aims to minimize the mutual information between the input $\boldsymbol{x}$ and the latent representation $\boldsymbol{z}$ while maximizing the mutual information between the latent representation and the output. The learning objective is formulated as:

$$\min I(X, Z) - \beta I(Z, Y) = \min \mathbb{E}_{(\boldsymbol{x}, \boldsymbol{z}) \sim \mathcal{P}(\boldsymbol{x}, \boldsymbol{z})} \log \frac{\mathcal{P}(\boldsymbol{z}|\boldsymbol{x})}{\mathcal{P}(\boldsymbol{z})} - \beta \mathbb{E}_{(y, \boldsymbol{z}) \sim \mathcal{P}(y, \boldsymbol{z})} \log \frac{\mathcal{P}(y|\boldsymbol{z})}{\mathcal{P}(y)} . \quad (2)$$

Since the true latent distributions of $\boldsymbol{x}, \boldsymbol{z}, y$ are unknown, we introduce variational distributions $\mathcal{Q}(\boldsymbol{z}|\boldsymbol{x})$ and $\mathcal{Q}(y|\boldsymbol{z})$ to the objective and have

$$Eq.(2) = \min \mathbb{E}_{(\boldsymbol{x},\boldsymbol{z})\sim\mathcal{P}(\boldsymbol{x},\boldsymbol{z})} \log \frac{\mathcal{P}(\boldsymbol{z}|\boldsymbol{x})\mathcal{Q}(\boldsymbol{z}|\boldsymbol{x})}{\mathcal{P}(\boldsymbol{z})\mathcal{Q}(\boldsymbol{z}|\boldsymbol{x})} - \beta \mathbb{E}_{(y,\boldsymbol{z})\sim\mathcal{P}(y,\boldsymbol{z})} \log \frac{\mathcal{P}(y|\boldsymbol{z})\mathcal{Q}(y|\boldsymbol{z})}{\mathcal{P}(y)\mathcal{Q}(y|\boldsymbol{z})}$$

$$= \min \iint_{(X,Z)} \mathcal{P}(\boldsymbol{z}|\boldsymbol{x})\mathcal{P}(\boldsymbol{x})[\log \frac{\mathcal{P}(\boldsymbol{z}|\boldsymbol{x})}{\mathcal{Q}(\boldsymbol{z}|\boldsymbol{x})} + \log \frac{\mathcal{Q}(\boldsymbol{z}|\boldsymbol{x})}{\mathcal{P}(\boldsymbol{z})}] dz d\boldsymbol{x}$$

$$- \beta \iint_{(Y,Z)} \mathcal{P}(y|\boldsymbol{z})\mathcal{P}(\boldsymbol{z})[\log \frac{\mathcal{P}(y|\boldsymbol{z})}{\mathcal{Q}(y|\boldsymbol{z})} + \log \frac{\mathcal{Q}(y|\boldsymbol{z})}{\mathcal{P}(y)}] dy dz \tag{3}$$

$$= \min \mathbb{E}_{\boldsymbol{x}\sim\mathcal{P}(\boldsymbol{x})} \mathrm{D}_{\mathrm{KL}}(\mathcal{P}(\boldsymbol{z}|\boldsymbol{x})\|\mathcal{Q}(\boldsymbol{z}|\boldsymbol{x})) + \mathbb{E}_{(\boldsymbol{x},\boldsymbol{z})\sim\mathcal{P}(\boldsymbol{x},\boldsymbol{z})} \log \frac{\mathcal{Q}(\boldsymbol{z}|\boldsymbol{x})}{\mathcal{P}(\boldsymbol{z})}$$

$$- \beta \mathbb{E}_{\boldsymbol{z}\sim\mathcal{P}(\boldsymbol{z})} \mathrm{D}_{\mathrm{KL}}(\mathcal{P}(y|\boldsymbol{z})\|\mathcal{Q}(y|\boldsymbol{z})) - \beta \mathbb{E}_{(y,\boldsymbol{z})\sim\mathcal{P}(y,\boldsymbol{z})} \log \frac{\mathcal{Q}(y|\boldsymbol{z})}{\mathcal{P}(y)},$$

where $\mathrm{D}_{\mathrm{KL}}(\cdot)$ is the KL-divergence. The term involving $\log \frac{\mathcal{Q}(\boldsymbol{z}|\boldsymbol{x})}{\mathcal{P}(\boldsymbol{z})}$ can be written as:

$$\mathbb{E}_{(\boldsymbol{x},\boldsymbol{z})\sim\mathcal{P}(\boldsymbol{x},\boldsymbol{z})} \log \frac{\mathcal{Q}(\boldsymbol{z}|\boldsymbol{x})}{\mathcal{P}(\boldsymbol{z})} = \iint_{(X,Z)} \mathcal{P}(\boldsymbol{z}|\boldsymbol{x})\mathcal{P}(\boldsymbol{x})[\log \mathcal{Q}(\boldsymbol{z}|\boldsymbol{x}) - \log \mathcal{P}(\boldsymbol{z})] dz d\boldsymbol{x}$$

$$= -\mathbb{E}_{\boldsymbol{x}\sim\mathcal{P}(\boldsymbol{x})}[\mathrm{H}_{\mathcal{P},\mathcal{Q}}(\boldsymbol{z}|\boldsymbol{x})] + \mathrm{H}(Z), \tag{4}$$

where

$$\mathrm{H}_{\mathcal{P},\mathcal{Q}}(\boldsymbol{z}|\boldsymbol{x}) = -\iint_{(X,Z)} \mathcal{P}(\boldsymbol{x})\mathcal{P}(\boldsymbol{z}|\boldsymbol{x}) \log \mathcal{Q}(\boldsymbol{z}|\boldsymbol{x}) dz d\boldsymbol{x}, \tag{5}$$

is the cross entropy between $\mathcal{P}(\boldsymbol{z}|\boldsymbol{x})$ and $\mathcal{Q}(\boldsymbol{z}|\boldsymbol{x})$, $\mathrm{H}(Z) = -\int_Z \mathcal{P}(\boldsymbol{z}) \log \mathcal{P}(\boldsymbol{z}) dz$ is the entropy of the random variable $Z$ under the probability distribution $\mathcal{P}(\boldsymbol{z})$. Similarly, we can derive:

$$\mathbb{E}_{(y,\boldsymbol{z})\sim\mathcal{P}(y,\boldsymbol{z})} \log \frac{\mathcal{Q}(y|\boldsymbol{z})}{\mathcal{P}(y)} = -\mathbb{E}_{\boldsymbol{z}\sim\mathcal{P}(\boldsymbol{z})}[\mathrm{H}_{\mathcal{P},\mathcal{Q}}(y|\boldsymbol{z})] + \mathrm{H}(Y). \tag{6}$$

Note that, $\mathrm{H}(Z)$ and $\mathrm{H}(Y)$ are constant and independent of our optimization problem. Therefore, we omit them from the objective. Substituting Eq. (4) and Eq. (6) into Eq. (3) yields:

$$Eq.(3) = \min \mathbb{E}_{\boldsymbol{x}\sim\mathcal{P}(\boldsymbol{x})} \mathrm{D}_{\mathrm{KL}}(\mathcal{P}(\boldsymbol{z}|\boldsymbol{x})\|\mathcal{Q}(\boldsymbol{z}|\boldsymbol{x})) - \mathbb{E}_{\boldsymbol{x}\sim\mathcal{P}(\boldsymbol{x})}[\mathrm{H}_{\mathcal{P},\mathcal{Q}}(\boldsymbol{z}|\boldsymbol{x})]$$

$$- \beta \mathbb{E}_{\boldsymbol{z}\sim\mathcal{P}(\boldsymbol{z})} \mathrm{D}_{\mathrm{KL}}(\mathcal{P}(y|\boldsymbol{z})\|\mathcal{Q}(y|\boldsymbol{z})) + \beta \mathbb{E}_{\boldsymbol{z}\sim\mathcal{P}(\boldsymbol{z})}[\mathrm{H}_{\mathcal{P},\mathcal{Q}}(y|\boldsymbol{z})]. \tag{7}$$

Here, we assume $p^e, p^d, q^e, q^d$ are discrete distributions, e.g., categorical over a finite codebook for $\boldsymbol{z}$ and over class labels for $y$. Under this assumption, the KL-divergence and cross entropy are nonnegative, hence:

$$Eq.\ (7) \le \mathbb{E}_{\boldsymbol{x}\sim\mathcal{P}(\boldsymbol{x})} \mathrm{D}_{\mathrm{KL}}(\mathcal{P}(\boldsymbol{z}|\boldsymbol{x})\|\mathcal{Q}(\boldsymbol{z}|\boldsymbol{x})) + \beta \mathbb{E}_{\boldsymbol{z}\sim\mathcal{P}(\boldsymbol{z})}[\mathrm{H}_{\mathcal{P},\mathcal{Q}}(y|\boldsymbol{z})]. \tag{8}$$

We therefore optimize the RHS as a tractable surrogate objective, which is an upper bound of the original optimization problem. Note that, optimizing this probability estimations can be approximated by introducing parametric encoder and decoder. Inspired by (Alemi et al., 2017), we approximate $\mathcal{P}(\boldsymbol{z}|\boldsymbol{x})$ and $\mathcal{P}(y|\boldsymbol{z})$ by $p(\boldsymbol{z}|\boldsymbol{x};\boldsymbol{\theta}_1)$, $p(y|\boldsymbol{z};\boldsymbol{\theta}_2)$ and turn to optimize $\boldsymbol{\theta}_1, \boldsymbol{\theta}_2$. For the introduced variational distributions $\mathcal{Q}(\boldsymbol{z}|\boldsymbol{x})$ and $\mathcal{Q}(y|\boldsymbol{z})$, we propose to approximate them by the global model $q(\boldsymbol{z}|\boldsymbol{x};\boldsymbol{\mu}_1), q(y|\boldsymbol{z};\boldsymbol{\mu}_2)$. The motivation is that the global model gathers the information of every client and thus becomes more accurate. It provides useful information in guiding the training of the client model (Yang et al., 2019; Ren et al., 2025). By applying this approximation, We derive the following learning objective for each client:

$$\min_{(\boldsymbol{\theta}_1,\boldsymbol{\theta}_2)} \mathbb{E}_{\boldsymbol{x}\sim\mathcal{P}(\boldsymbol{x})} \mathrm{D}_{\mathrm{KL}}(p^e(\boldsymbol{z}|\boldsymbol{x};\boldsymbol{\theta}_1)\|q^e(\boldsymbol{z}|\boldsymbol{x};\boldsymbol{\mu}_1)) + \beta \mathbb{E}_{\boldsymbol{z}\sim\mathcal{P}(\boldsymbol{z})}[\mathrm{H}_{p^d,q^d}(y|\boldsymbol{z};\boldsymbol{\theta}_2,\boldsymbol{\mu}_2)]. \tag{9}$$

The client subscript has been omitted for simplicity of notation since the clients share the objective in horizontal FL. Note that Eq. (9) encourages the local model to produce intermediate and final outputs similar to those of the global model, which aligns with established practices in the federated learning literature for training local models (Li et al., 2020; Collins et al., 2021).

Next, we propose a minimax objective for data selection. Minimax is a common technique in active learning (Steven et al., 2008; Huang et al., 2014; Ghafarian & Yazdi, 2019), which selects the data

---

**Algorithm 1** Fed-MADS

---

**Input:** Labeled set $\mathbb{L}_i$, unlabeled set $\mathbb{U}_i$ for each client $i \in \{1, \ldots, k\}$, query budget per round $b$, tradeoff $\beta$, global model $f_g = \{q^e(\boldsymbol{z}|\boldsymbol{x}; \boldsymbol{\mu}_1), q^d(y|\boldsymbol{z}; \boldsymbol{\mu}_2)\}$.

1: **for** each communication round **do**
2:      **for** each client $i$ **in parallel do**
3:          Receive global model $f_g$ from server
4:          Train local model $f_i = \{p_i^e(\boldsymbol{z}|\boldsymbol{x}; \boldsymbol{\theta}_1), p_i^d(y|\boldsymbol{z}; \boldsymbol{\theta}_2)\}$ using current $\mathbb{L}_i$ and global model
5:          **for** each $\boldsymbol{x} \in \mathbb{U}_i$ **do**
6:              Compute KL-divergence between the intermediate outputs of local and global models:

$$s_1 = \mathrm{D}_{\mathrm{KL}}\left(p_i^e(\boldsymbol{z}|\boldsymbol{x}; \boldsymbol{\theta}_1)\|q^e(\boldsymbol{z}|\boldsymbol{x}; \boldsymbol{\mu}_1)\right)$$

7:              Compute cross-entropy between the final prediction of local and global models:

$$s_2 = \mathrm{H}_{f^i, f_g}\left(y|\boldsymbol{x}; \boldsymbol{\theta}_1, \boldsymbol{\theta}_2, \boldsymbol{\mu}_1, \boldsymbol{\mu}_2\right)$$

8:              Calculate the final selection score: $\mathrm{Score}(\boldsymbol{x}) = s_1 + \beta s_2$
9:          **end for**
10:          Select $\mathbb{Q}_i \subset \mathbb{U}_i$, $|\mathbb{Q}_i| = b$ with top-$b$ scores
11:          Query labels for $\mathbb{Q}_i$, update: $\mathbb{L}_i \leftarrow \mathbb{L}_i \cup \mathbb{Q}_i$, $\mathbb{U}_i \leftarrow \mathbb{U}_i \setminus \mathbb{Q}_i$
12:          Retrain local model $f_i$ on updated $\mathbb{L}_i$
13:          Send model parameters $\boldsymbol{\theta}_1, \boldsymbol{\theta}_2$ to server
14:      **end for**
15:      Server aggregates local models to update global model $\boldsymbol{\mu}_1, \boldsymbol{\mu}_2$
16: **end for**

---

points that potentially cause the largest increment in the objective function. More specifically, in model training, the objective is generally formulated as a minimization form, whereas in active data selection, the goal is to identify samples that with larger losses (i.e., those would maximize the objective function, which are considered more informative). By adopting the minimax form, we unify the learning and data selection procedures, making the overall pipeline more consistent and logical.

To formulate and solve the minimax problem, recall that we are considering pool-based AL setting, i.e, there is a large pool of unlabeled data $\mathbb{U}$ in which each element is sampled i.i.d. from the latent distribution $\mathcal{P}(\boldsymbol{x})$. Therefore, we use the Monte Carlo method to approximate the expectation in the first term in Eq. (9). For the second expectation, it requires samples from the marginal distribution $\mathcal{P}(\boldsymbol{z})$, which is not available. Here, we propose to approximate it using the samples of $\boldsymbol{x}$, since $\boldsymbol{z}$ is dependent on $\boldsymbol{x}$, i.e.,

$$\mathbb{E}_{\boldsymbol{z} \sim \mathcal{P}(\boldsymbol{z})}[\mathrm{H}_{p^d, q^d}(y|\boldsymbol{z}; \boldsymbol{\theta}_2, \boldsymbol{\mu}_2)] \tag{10}$$

$$\approx \mathbb{E}_{\boldsymbol{x} \sim \mathcal{P}(\boldsymbol{x})}\mathbb{E}_{\boldsymbol{z} \sim p^e(\boldsymbol{z}|\boldsymbol{x}; \boldsymbol{\theta}_1)}[\mathrm{H}_{p^d, q^d}(y|\boldsymbol{z}; \boldsymbol{\theta}_2, \boldsymbol{\mu}_2)] \tag{11}$$

$$\approx \mathbb{E}_{\boldsymbol{x} \sim \mathcal{P}(\boldsymbol{x})}\, \mathrm{H}_{f, f_g}(y|\boldsymbol{x}; \boldsymbol{\theta}_1, \boldsymbol{\theta}_2, \boldsymbol{\mu}_1, \boldsymbol{\mu}_2))\ , \tag{12}$$

where $f(y|\boldsymbol{x}; \boldsymbol{\theta}_1, \boldsymbol{\theta}_2)$ is the prediction of the local model given the input $\boldsymbol{x}$, $f_g(y|\boldsymbol{x}; \boldsymbol{\mu}_1, \boldsymbol{\mu}_2)$ is the output of the global model. Again, by using the Monte Carlo method to approximate this surrogate, we derive the following objective function

$$\min_{(\boldsymbol{\theta}_1, \boldsymbol{\theta}_2)} \max_{\mathbb{Q} \subseteq \mathbb{U}, |\mathbb{Q}|=b} \frac{1}{|\mathbb{Q}|} \sum_{\boldsymbol{x} \in \mathbb{Q}}[\mathrm{D}_{\mathrm{KL}}(p^e(\boldsymbol{z}|\boldsymbol{x}; \boldsymbol{\theta}_1)\|q^e(\boldsymbol{z}|\boldsymbol{x}; \boldsymbol{\mu}_1))] + \beta \frac{1}{|\mathbb{Q}|} \sum_{\boldsymbol{x} \in \mathbb{Q}} \mathrm{H}_{f, f_g}(y|\boldsymbol{x}; \boldsymbol{\theta}_1, \boldsymbol{\theta}_2, \boldsymbol{\mu}_1, \boldsymbol{\mu}_2)\ . \tag{13}$$

The first term in Eq. (13) estimates the KL-divergence between the local and the global models' latent representations, while the second term measures the cross entropy with respect to label prediction of both models. Therefore, Eq. (13) suggests that the data point that incurs a large divergence between local and global models in terms of both latent representations and label predictions should be selected for querying.

In our algorithmic implementation, we estimate the first term by calculating the KL-divergence between the intermediate outputs of client and global models. For the second term, we calculate the cross entropy between the final outputs of both models. Note that, we update the model following the standard approach used in XFL; in other words, we use Eq. (13) solely for data selection.

While Eq. (13) could also be used to regularize the training of the local model, we refrain from modifying the training objective. This is because most existing FL schemes already incorporate similar regularization (Li et al., 2020; Durmus et al., 2021; Shi et al., 2023), and maintaining the original training method allows our approach to remain more generally applicable. The main steps of the proposed algorithm `Fed-MADS` is summarized at Algorithm 1.

### 3.4 ANALYSIS

We analyze the communication cost and computation cost of `Fed-MADS` in the following. `Fed-MADS` is designed to be communication-efficient and privacy-preserving. Since the proposed framework does not change the training process of model, the computation cost of training the local model is the same to that of existing XFL methods. In each round, the clients conduct data selection and labeling locally. Therefore, the data selection process does not incur any communication cost. The computation cost of `Fed-MADS` is mainly incurred by data selection. The data selection process requires computing the cross-entropy and KL-divergence between the local and global models' outputs, which can be efficiently implemented using matrix operations. This procedure scales linearly with the number of unlabeled data points, i.e., $O(|\mathbb{U}_i|)$, which is highly efficient. Therefore, `Fed-MADS` is communication-efficient and computationally efficient, making it suitable for practical applications in explainable FAL.

## 4 EXPERIMENTAL EVALUATION

### 4.1 EXPERIMENT SETTINGS

**Target Model.**   We adopt the state-of-the-art XFL framework LR-XFL (Zhang & Yu, 2024) as our base model, and conduct active learning to learn this model with possibly minimum label querying. LR-XFL learns a semantically-rich representation along with the prediction of the labels. The representation is used to form logical explanations for the predictions, allowing users to understand how it arrives at its outputs. Our experimental setup closely follows the empirical settings in LR-XFL's GitHub project (Yanci87, 2025), including datasets, model architectures, hyperparameters, performance metrics, etc. We refer to the source code of LR-XFL for the details of model learning.

**Datasets.**   We conduct our experiments on 4 cross-domain benchmark datasets: **MNIST (Even/Odd)** (LeCun et al., 1998), **MIMIC-II** (Saeed et al., 2011), **V-Dem County-Year** (Coppedge et al., 2022) and **Credit Card** (Dal Pozzolo et al., 2015). MNIST is a well-known handwritten character recognition dataset. We follow the empirical setting in (Zhang & Yu, 2024) to transform it into a binary classification problem. After applying the same data augmentation procedure in LR-XFL, there are 120000 instances in total. MIMIC-II is a medical dataset, whose task is to predict recovering or dying patients after ICU admission. V-Dem dataset contains the detailed democracy ratings over 200 countries. The learning target is distinguishing electoral democracies from non-electoral ones. Credit card dataset consists of transaction records made by European cardholders in September 2013. There are 284807 instances and the learning goal is to detect frauds.

**FAL Settings.**   The FAL system has 10 clients. Each client is equipped with an ideal oracle capable of providing accurate labels for unlabeled data at a fixed cost. For each dataset, 70% of its data is randomly sampled as training set. Half of the rest is used as the validation set (which is required by LR-XFL for model training), and the remaining half is used as the test set. The training set is further divided into 10 clients uniformly and evenly, each with a unique subset of data. Within each client, 5% of the local data is randomly selected to form the initial labeled set, while the remaining 95% serves as the unlabeled pool for querying. In each FAL round, clients receive the updated global model from the central server and select 5 unlabeled samples from their local pool based on a predefined query strategy. These samples are then labeled by the local oracle. Clients subsequently train their local models on the updated labeled datasets and transmit the resulting model parameters to the server for aggregation. This iterative process continues until the total query budget is depleted or model performance converges.

**Comparison Methods.**   We compare `Fed-MADS` with the following baselines and the state-of-the-art FAL methods:

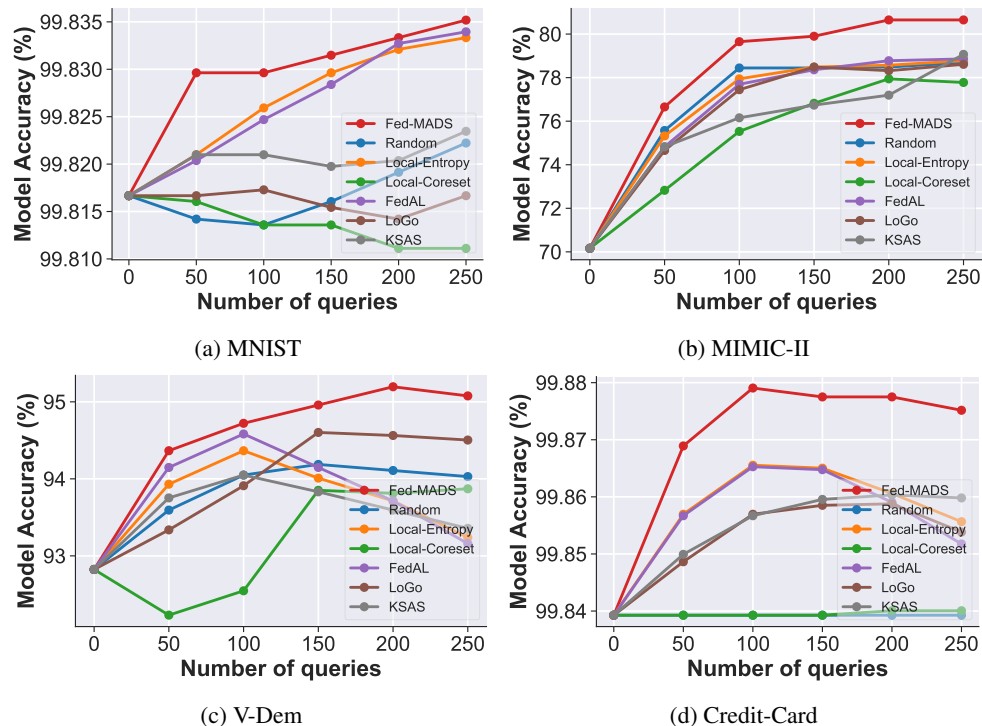

Figure 2: The learning curves of compared method on 4 benchmark datasets. The performance metric is model accuracy (%). (a) MNIST; (b) MIMIC-II; (c) V-Dem; (d) Credit-Card.

- **Random**: Uniformly select samples from the unlabeled pool.
- **Local-Entropy** (Ahmed et al., 2023): Select informative samples based on the entropy of local model predictions.
- **Local-Coreset** (Sener & Savarese, 2018): Select representative samples based on the Coreset selection method.
- **LoGo** (Kim et al., 2023): Select informative and diverse data points based on clustering and prediction entropy.
- **KSAS** (Cao et al., 2023): Select informative samples by estimating the KL-divergences between the class-weighted predictions of local and global models.
- **FedAL** (Deng et al., 2022): Select informative samples by the entropy of mean output of local and global models.

All hyperparameters of the compared methods are set as the suggested values in the original paper. For the trade-off $\beta$ in `Fed-MADS`, we select it from $\{0.1, 1, 10, 100\}$ using the validation set.

**Performance Metrics.** We follow the XFL literature (Zhang & Yu, 2024) to use 3 metrics to evaluate the performance of the model, i.e., *model accuracy*, *rule accuracy* and *rule fidelity*.

- **Model accuracy** is calculated as the percentage of correctly predicted labels on the test set.
- **Rule accuracy** estimates the consistency between the rule predictions and the ground truth labels. It is calculated on a given class $c$. Specifically, denote by $\mathbb{T}$ and $\mathbb{T}^c$ the test set and the subset of the test set in class $c$, respectively. Let $|\cdot|$ be the size of a set. Suppose there are $m_1$ data points among $\mathbb{T}^c$ that satisfy the propositions in the rule of class $c$ generated by the model, and $m_2$ data points among $\mathbb{T} \setminus \mathbb{T}^c$ that do not satisfy the rule of class $c$. The rule accuracy is defined as: $(m_1 + m_2)/|\mathbb{T}|$.
- **Rule fidelity** is defined in a similar way as rule accuracy, but it estimates the consistency between the rule predictions and the model predicted labels. It can be calculated by simply replacing the ground truth labels with the model predicted labels in the above definition of rule accuracy.

Table 1: Results of rule accuracy (%) and rule fidelity (%) of compared methods on 4 benchmark datasets. The mean and standard deviation values of each learning curve are reported. The best performances are highlighted in boldface.

| Methods | Datasets | | | |
|---|---|---|---|---|
| | MNIST | MIMIC-II | V-Dem | Credit-Card |
| Rule Accuracy | | | | |
| Fed-MADS | **$92.956 \pm 2.289$** | **$56.516 \pm 6.033$** | **$89.131 \pm 6.265$** | **$58.593 \pm 3.218$** |
| Random | $90.387 \pm 0.610$ | $49.410 \pm 2.930$ | $81.328 \pm 2.933$ | $49.368 \pm 2.361$ |
| Local-Entropy | $83.988 \pm 4.135$ | $50.075 \pm 3.223$ | $84.895 \pm 5.009$ | $55.288 \pm 2.673$ |
| Local-Coreset | $83.844 \pm 6.084$ | $49.944 \pm 3.059$ | $81.109 \pm 4.202$ | $44.245 \pm 4.777$ |
| FedAL | $85.947 \pm 1.928$ | $49.648 \pm 3.555$ | $83.995 \pm 4.616$ | $55.603 \pm 2.811$ |
| LoGo | $84.943 \pm 2.550$ | $48.983 \pm 2.602$ | $81.748 \pm 2.871$ | $55.926 \pm 2.003$ |
| KSAS | $87.842 \pm 3.751$ | $51.408 \pm 4.070$ | $85.420 \pm 5.210$ | $53.807 \pm 1.652$ |
| Rule Fidelity | | | | |
| Fed-MADS | **$93.378 \pm 2.130$** | **$73.960 \pm 4.840$** | **$93.110 \pm 6.908$** | **$97.201 \pm 6.214$** |
| Random | $91.023 \pm 0.534$ | $66.289 \pm 1.821$ | $84.025 \pm 2.974$ | $79.603 \pm 4.140$ |
| Local-Entropy | $85.336 \pm 3.429$ | $66.278 \pm 3.351$ | $88.390 \pm 5.616$ | $94.420 \pm 6.415$ |
| Local-Coreset | $85.259 \pm 5.219$ | $67.863 \pm 2.645$ | $84.203 \pm 4.786$ | $70.347 \pm 8.279$ |
| FedAL | $87.467 \pm 1.496$ | $66.687 \pm 2.768$ | $87.495 \pm 5.279$ | $94.419 \pm 6.415$ |
| LoGo | $85.951 \pm 2.349$ | $65.113 \pm 1.126$ | $84.460 \pm 2.931$ | $96.275 \pm 6.145$ |
| KSAS | $88.961 \pm 3.484$ | $69.829 \pm 3.929$ | $89.517 \pm 6.067$ | $96.271 \pm 6.142$ |

To compare the performance of active data selection methods, we utilize learning curves of different selection methods as the metric. The learning curves are generated by plotting the model performance against the labeling cost made by each method. The x-axis represents the number of queries or cost of labeling data, while the y-axis represents the model performance. For the numerical results, we also employ mean value of the learning curve to quantify the overall performance. This metric is proportional to the area under the curve (AUC) and a larger value indicates a better performance.

## 4.2 COMPARISON RESULTS AND DISCUSSION

We plot the learning curves of the compared methods on the 4 datasets in Figure 2. The results demonstrate that Fed-MADS consistently outperforms the other methods across all datasets. In particular, Fed-MADS achieves the highest performance with the fewest queries, indicating the effectiveness of considering the implicit training dynamic using IB principle in data selection for XFL model. The Local-Coreset method performs less effectively in the FL setting, likely due to its neglect of data informativeness when selecting samples. Although the Random method performs competitively on the MIMIC-II dataset, it fails to generalize to the other datasets. We conjecture that this is because MIMIC-II exhibits more pronounced distribution shifts, which inadvertently favor random sampling. In contrast, Local-Coreset prioritizes coverage of the data space rather than mitigating distribution shift, leading to inconsistent performance compared to the Random baseline. Local-Entropy and FedAL perform well in most cases, indicating the effectiveness of considering the prediction uncertainty in data selection. However, they are less effective than Fed-MADS. This phenomenon can be attributed to the fact that they do not consider the training dynamics of the model, which is crucial for selecting informative samples in XFL setting. The performances of LoGo and KSAS vary across datasets. This variability may stem from LoGo's reliance on clustering strategies, which can be dataset-dependent and may not always capture sample informativeness effectively. KSAS, on the other hand, selects samples based on the mean outputs of local and global models, which may lead to suboptimal choices, particularly when either model is undertrained and thus fails to provide reliable guidance.

We further report model performance in terms of *rule accuracy* and *rule fidelity* in Table 1. These two metrics are essential in evaluating the explainability of FL models. As shown in the table, Fed-MADS consistently outperforms other methods across all datasets with respect to both rule accuracy and rule fidelity. This demonstrates the effectiveness of our method in selecting informative samples for XFL models, thereby enhancing the rule learning capabilities. We attribute this advantage to the design of Fed-MADS, which takes into account the training dynamics of the model and incorporates the prediction divergence from intermediate model outputs. In contrast, the results of the other compared methods reveal that the performance varies significantly across datasets. For instance, the KSAS method performs well on MIMIC-II but fails on other datasets. This suggests that relying solely on

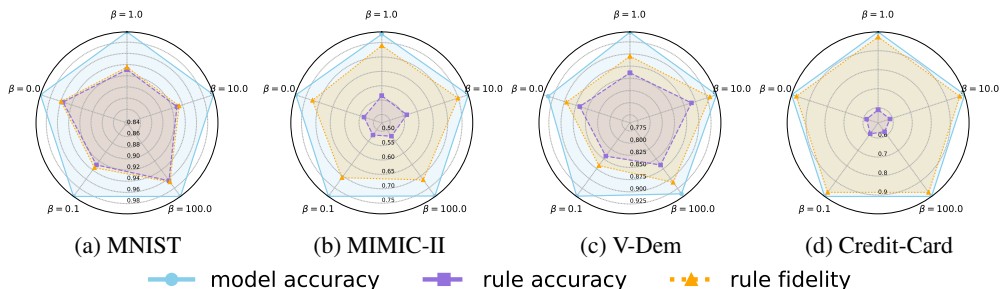

(a) MNIST  (b) MIMIC-II  (c) V-Dem  (d) Credit-Card

model accuracy  rule accuracy  rule fidelity

Figure 3: Parameter sensitivity of `Fed-MADS` on 4 benchmark datasets. The value of $\beta$ varies from $\{0, 0.1, 1, 10, 100\}$. The performance metrics include model accuracy, rule accuracy and rule fidelity. (a) MNIST; (b) MIMIC-II; (c) V-Dem; (d) Credit-Card.

traditional data selection criteria may result in suboptimal performance. Therefore, incorporating training dynamics is crucial for improving model explainability in this context.

### 4.3 ABLATION STUDY ON THE IMPACT OF THE TRADE-OFF PARAMETER

To examine the sensitivity of `Fed-MADS` to the trade-off parameter $\beta$, we conduct experiments on 4 benchmark datasets by varying $\beta \in \{0, 0.1, 1, 10, 100\}$. The special case of $\beta = 0$ serves as a degenerated variant of `Fed-MADS`, which omits the divergence in label predictions and relies solely on the discrepancy in hidden representations. This setting provides an ablation study to evaluate the individual contribution of the prediction divergence term.

The results are illustrated in Figure 3, where we report three performance metrics: model accuracy, rule accuracy, and rule fidelity. Overall, `Fed-MADS` demonstrates strong robustness across a wide range of $\beta$ values. Nevertheless, a general trend emerges: higher values of $\beta$ tend to improve rule accuracy and rule fidelity across most datasets, suggesting that incorporating the divergence in label predictions between local and global models enhances the selection of informative samples. This observation reinforces the intuition that prediction disagreement is a valuable signal for improving explainability in federated learning. Specifically, for the MIMIC-II and Credit-Card datasets, the performance gains with increasing $\beta$ are more pronounced, especially in rule fidelity. This indicates that in complex or imbalanced domains, prediction divergence plays an even more critical role. In contrast, the MNIST dataset shows relatively stable performance across all $\beta$ values, likely due to its lower complexity and more balanced distribution. Interestingly, even with $\beta = 0$, `Fed-MADS` achieves competitive results, particularly in terms of rule accuracy and rule fidelity. This highlights the effectiveness of leveraging latent representation discrepancies alone, and suggests that hidden-layer divergence is a strong proxy for data informativeness in XFL settings.

As a conclusion, while `Fed-MADS` performs robustly without prediction divergence, incorporating it through a moderate-to-large $\beta$ leads to consistently improved explainability, particularly in complex real-world datasets.

## 5 CONCLUSIONS AND FUTURE WORK

In this paper, we present `Fed-MADS`, a novel framework for explainable FAL. Inspired by the IB principle, `Fed-MADS` employs a theoretical lens to analyze the training dynamics of the local and global explainable models. By formulating a minimax AL objective derived from the IB principle, our method efficiently selects the most informative unlabeled samples across federated clients. The proposed query method integrates both global and local models in data selection by implementing the variational distributions using local and global parametric models. Thus, it naturally accords with the FL setting. Extensive experiments on 4 benchmark datasets demonstrated that `Fed-MADS` consistently outperformed state-of-the-art FAL methods in terms of model accuracy, rule accuracy, and rule fidelity. However, we acknowledge that `Fed-MADS` has limitations, e.g., it relies on the assumption that the global model is well-trained during the data selection phase. This might not hold in scenarios with limited communication rounds or when the global model is not sufficiently robust. Future work could explore strategies to address this limitation and enhance the performance.

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

## A   LLM USAGE STATEMENT

Large Language Models (LLMs) were employed to assist in the preparation of this manuscript. Specifically, the authors used LLMs to polish the wording and sentence structure for improved clarity and readability. In addition, LLMs were applied to check spelling, as well as to identify and correct minor errors in symbols and formulas. All substantive ideas, analyses, and conclusions presented in this paper are solely those of the authors.

## B   PSEUDO-CODE

We present the core python codes of `Fed-MADS` as follows. A more refined version with detailed documentation will be released publicly on GitHub at a later stage.

```python
import torch.nn.functional as F

# collect features and scores
feats, q1_list, q2_list = [], [], []
for data, _ in unlab:
    x = data.to(cli_model.device)
    cmid, cpred = cli_model.get_mid_and_final_output(x)
    fmid, fpred = glob_model.get_mid_and_final_output(x)

    # q1: KL-divergence between flattened cmid and fmid
    fc = cmid.flatten(1)
    ff = fmid.flatten(1)
    log_p = F.log_softmax(fc, dim=1)
    q = F.softmax(ff, dim=1)
    kl = F.kl_div(log_p, q, reduction='none').sum(1)

    # q2: CE between cfinal and ffinal
    p = F.softmax(cpred, dim=1)
    log_q = F.log_softmax(fpred, dim=1)
    ce = -(p * log_q).sum(1)

    q1_list.append(kl.cpu())
    q2_list.append(ce.cpu())
```

```
648  q1s  = torch.cat(q1_list, 0)
649  q2s  = torch.cat(q2_list, 0)
650  final_scores = q1s + q_coef * q2s
```

**Software:** All experiments are implemented using PyTorch 2.1.2 and Python 3.11. The CUDA version is 12.4, pytorch-lightning version is 1.9.5.

**Hardware:** Experiments are conducted on a private computing server equipped with AMD Ryzen Threadripper PRO 5965WX 24-Cores, 3 NVIDIA RTX A5000 graphic cards, and 184GB of RAM.

## C ADDITIONAL EXPERIMENTAL RESULTS

This section presents the performance results of the compared methods under varying sizes of the initially labeled dataset. Specifically, we vary the proportion of initially labeled data among 10%, 15%, 20%, and evaluate the performance of different compared methods accordingly. We report the mean values of the learning curves for each method, considering three performance metrics: model accuracy, rule fidelity, and rule accuracy.

The results corresponding to the initial labeled set sizes of 10%, 15%, and 20% are shown in Figure 4, Figure 5, and Figure 6, respectively. As observed, our proposed method consistently outperforms the baselines in most scenarios. Notably, the performance of Fed-MADS improves as the size of the initial labeled set increases. A plausible explanation is that a larger initial labeled set yields a more robust and well-trained global model, enabling Fed-MADS to capitalize more effectively on the global model's enhanced quality.

Furthermore, Fed-MADS demonstrates a particularly significant advantage in terms of rule fidelity and rule accuracy. We attribute this to the effectiveness of our proposed query strategy, which appears to better support XFL models in achieving higher explainability.

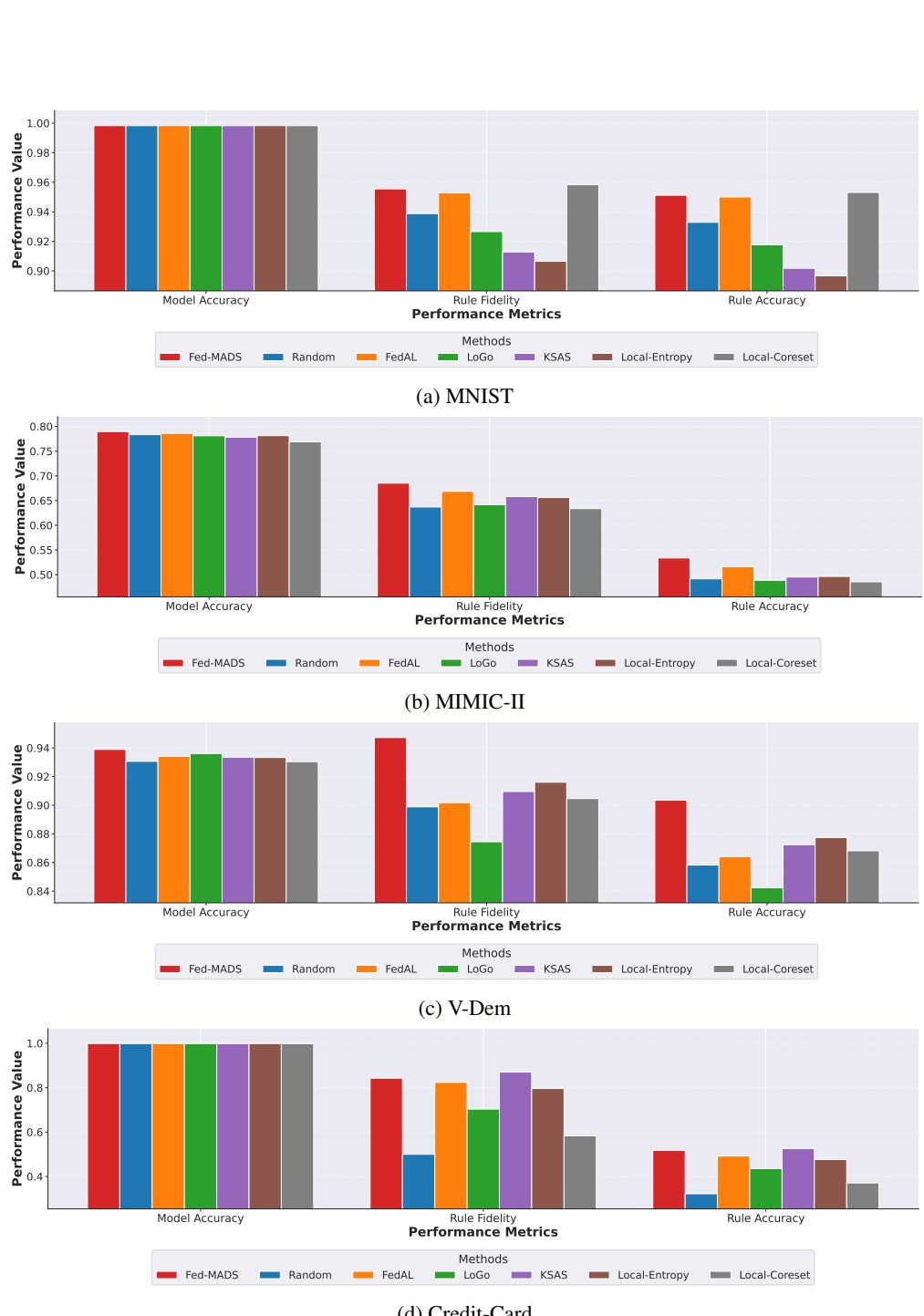

(a) MNIST

(b) MIMIC-II

(c) V-Dem

(d) Credit-Card

Figure 4: The performance comparison results with initially labeled size of 10%.

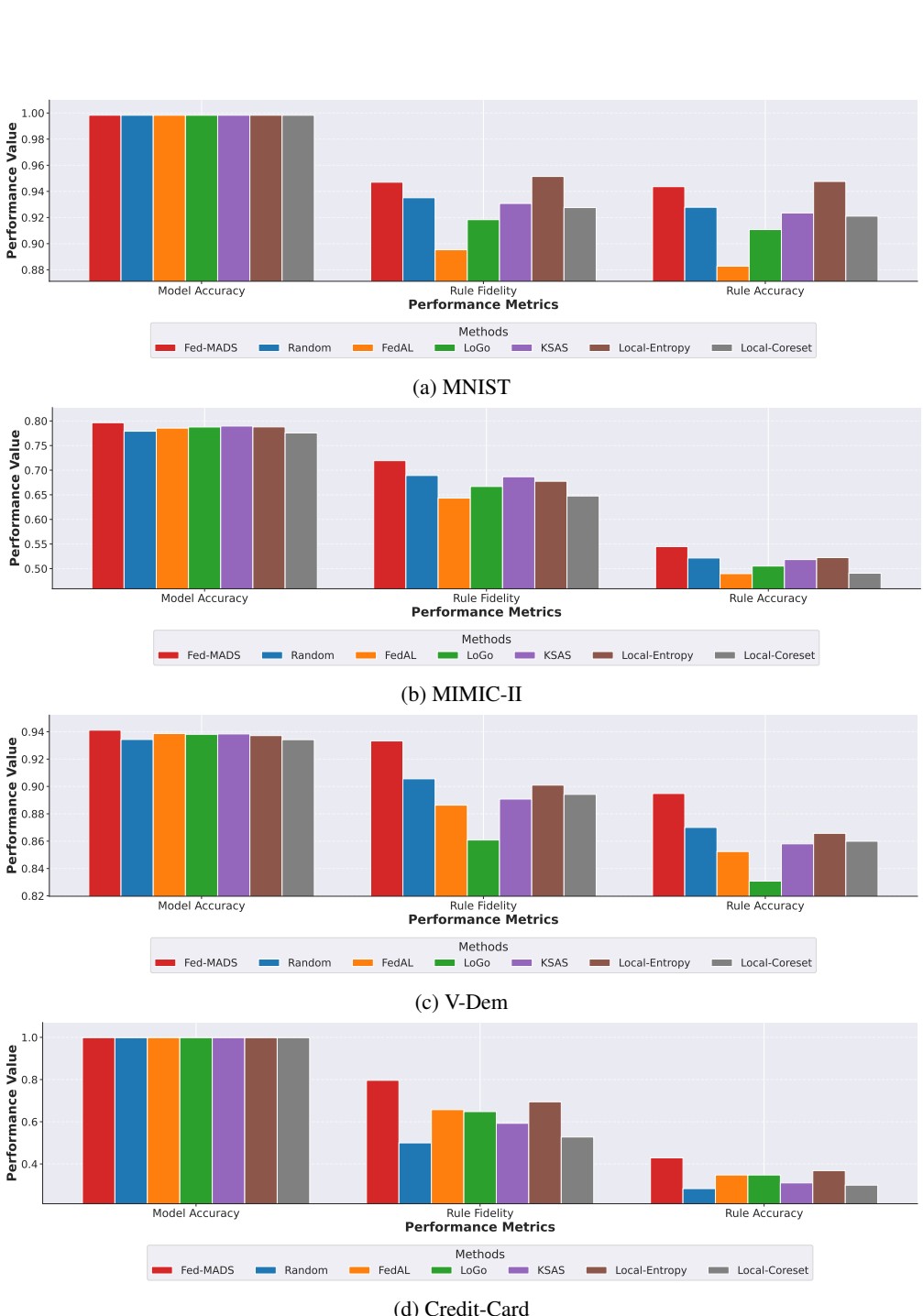

(a) MNIST

(b) MIMIC-II

(c) V-Dem

(d) Credit-Card

Figure 5: The performance comparison results with initially labeled size of 15%.

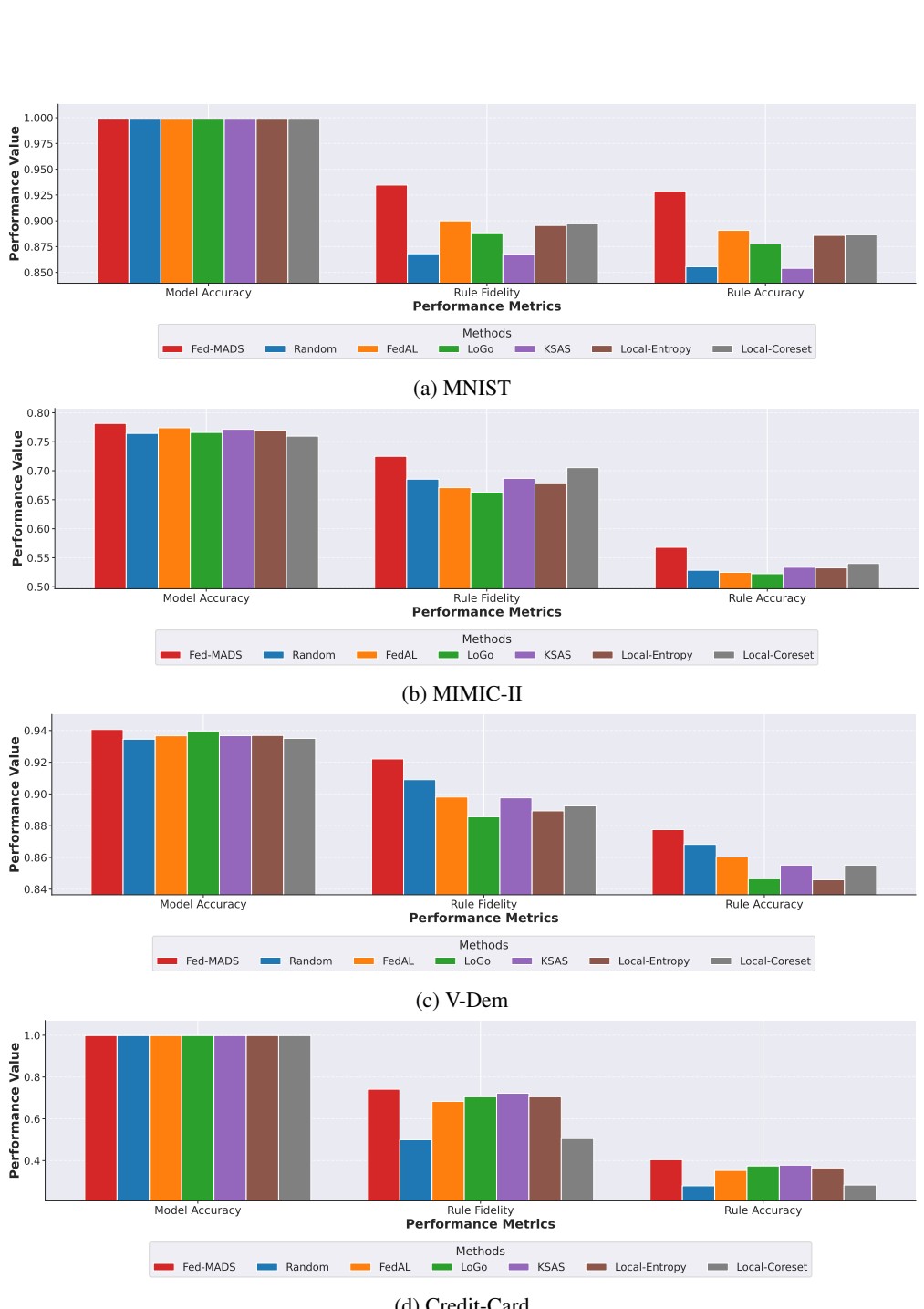

Figure 6: The performance comparison results with initially labeled size of 20%.

