# OpenReview forum: "Information Bottleneck-inspired Efficient and Explainable Federated Active Learning"
_ICLR.cc/2026/Conference — Submitted to ICLR 2026_

### Official Review · Reviewer_ieCM · 2025-10-29

**Soundness:** 2
**Presentation:** 3
**Contribution:** 2
**Rating:** 2
**Confidence:** 4

**Summary:**

The paper proposes a federated active learning (FAL) method derived from the information bottleneck, selecting unlabeled samples exhibiting significant divergence between local and global models in both latent representations and predicted labels. The four datasets demonstrate that our method significantly outperforms state-of-the-art FAL approaches, achieving superior performance with fewer labeled data points.

**Strengths:**

1. This paper presents a clearly written narrative and well-structured exposition that is easy to follow.
2. This paper proposes a FAL method grounded in the IB, selecting unlabeled samples that show large divergences between local and global models in both latent representations and predicted labels.

**Weaknesses:**

1. Lacks discussion and head-to-head comparisons with the latest FAL baselines, including TKDE 2025-CHASe and IJCAI 2025-IFAL; the related-work and experimental sections should position against these methods.
- TKDE-2025-CHASe: Client Heterogeneity-Aware Data Selection for Effective Federated Active Learning
- IJCAI-2025-Inconsistency-Based Federated Active Learning

2. Uses a highly limited experimental setup (mostly IID and small scale) and lacks systematic studies along key considerations that matter in FAL: client heterogeneity, number of clients, and model/dataset scales.

3. Omits a cost analysis of the data-selection module, with no reporting of time and memory overheads. For a practical FAL system, end-to-end and per-round costs of selection are necessary.

4. While grounded in the information bottleneck, the paper provides insufficient qualitative/quantitative analyses explaining why the IB-based criterion selects effective samples. Deeper probing would strengthen interpretability and insights; without it, the work reads more like an application and is less compelling for the community.

**Questions:**

Please refer to the weaknesses.

---

### Official Review · Reviewer_8SPU · 2025-10-31

**Soundness:** 3
**Presentation:** 3
**Contribution:** 3
**Rating:** 4
**Confidence:** 4

**Summary:**

This paper proposes Fed-MADS, a novel federated active learning (FAL) framework designed for explainable federated learning (XFL). The core problem addressed is the high cost of data labeling required to train effective XFL models. The authors leverage the Information Bottleneck (IB) principle to model the training dynamics between local and global models. From the IB objective, they derive a minimax data selection strategy. This strategy prioritizes querying unlabeled samples that exhibit the largest divergence between the local and global models, measured in terms of both their latent representations (via KL-divergence) and their final predictions (via cross-entropy). The paper claims that this approach leads to significant performance gains in model accuracy and explainability metrics (rule accuracy, rule fidelity) with fewer labeled samples, outperforming several state-of-the-art FAL methods on four benchmark datasets.

**Strengths:**

1. The primary strength of this paper is the novel application of the Information Bottleneck (IB) principle to the problem of active data selection in a federated setting. While IB has been used to analyze deep learning and federated learning, its use to derive a concrete query strategy for FAL is new and provides a fresh theoretical perspective that moves beyond conventional uncertainty-based methods.

2. The paper is generally well-written and easy to follow. The overall framework is clearly depicted in Figure 1, and the proposed algorithm is concisely summarized in Algorithm 1. The motivation for the work is well-articulated in the introduction.

3. The problem of data-efficient training for explainable models in a privacy-preserving context is highly significant. If the claims hold, this work would represent a valuable contribution by providing a principled and effective method to reduce labeling costs in XFL, making it more practical for real-world applications.

**Weaknesses:**

1. The claim that the selection strategy is "derived from the IB principle" is overstated. The derivation involves a critical approximation step that lacks rigorous justification. Specifically, the transition from Eq. (10) to Eq. (11) $E_{z\sim P(z)}[H_{p_d,q_d}(y|z;\theta_2,\mu_2)]\approx E_{x\sim P(x)}E_{z\sim p_e(z|x;\theta_1)}[H_{p_d,q_d}(y|z;\theta_2,\mu_2)]$ is not theoretically sound without strong, unstated assumptions. The expectation over the marginal distribution $P(z)$ is replaced by an expectation over the empirical data distribution $P(x)$ and the local model's encoder. This seems to be a heuristic choice made for computational convenience to arrive at a sample-wise score, rather than a principled derivation. This leap of faith undermines the theoretical foundation of the entire method, making it appear more as a well-motivated heuristic with a post-hoc IB justification.

2. The final selection score, $Score(x)=DKL(latent)~+\beta * CE(prediction)$, essentially boils down to selecting samples where the local (student) and global (teacher) models disagree. This "disagreement-based" querying is a classic and well-explored idea in the active learning literature (e.g., query-by-committee). Methods like KSAS also explicitly use the divergence between local and global model predictions. The paper fails to adequately differentiate its core mechanism from these existing ideas, beyond the IB framing. Given the weak derivation, the contribution risks being perceived as an incremental variant of disagreement-based sampling.

3. The paper explicitly states that the study "focuses on horizontal FL scenarios with i.i.d. data" (Line 118). This is a major limitation that severely curtails the significance and generalizability of the results. Data heterogeneity (non-i.i.d. data) is a fundamental and defining challenge in federated learning. In a non-i.i.d. setting, a large divergence between a local and global model could simply be due to the client's local data distribution shift, rather than the sample being genuinely informative for the global task. The proposed Fed-MADS method might be systematically biased towards selecting samples from clients with unique data distributions, which may not be optimal for improving the global model. Ignoring the non-i.i.d. case makes the evaluation feel incomplete and unrealistic for the FL domain.

**Questions:**

1. Could the authors provide a more rigorous justification for the approximation made between Eq. (10) and Eq. (11)? What are the underlying assumptions required for this approximation to hold, and how do they affect the validity of the final objective? Without this, the connection to the IB principle feels tenuous.

2. The core idea of selecting samples based on local-global model disagreement is shared with other methods. For instance, KSAS also measures the KL-divergence between local and global predictions. Could you please elaborate on the fundamental difference between Fed-MADS and other disagreement-based FAL methods? Why is the inclusion of the latent representation divergence (the $s_1$ term) critical, and why is the IB formulation superior to a more direct heuristic formulation of model disagreement?

3. The experiments are conducted exclusively on i.i.d. data. This is a critical omission for a paper in federated learning. How do you expect Fed-MADS to perform in a non-i.i.d. setting? Is there a risk that the selection score would be dominated by distribution shift rather than sample informativeness, potentially leading to suboptimal or biased data selection? A discussion or, ideally, preliminary experiments on non-i.i.d. data would be necessary to assess the practical viability of the method.

---

### Official Review · Reviewer_cVHf · 2025-11-01

**Soundness:** 2
**Presentation:** 2
**Contribution:** 2
**Rating:** 2
**Confidence:** 4

**Summary:**

This paper proposes a simple way to choose which data should be labeled in Federated Learning, where many users train a model together without sharing their raw data. The key idea is to compare how a local model and a global shared model disagree on each unlabeled sample, both in their final prediction and in their internal feature representation. Samples with the biggest disagreement are chosen for labeling. The authors motivate this idea using the information bottleneck principle and test it on several datasets, showing that it improves accuracy and the quality of model explanations while using fewer labels.

**Strengths:**

This paper takes well-known ideas from active learning and federated learning and combines them, using the information-bottleneck view to explain why the approach makes sense. The method is simple, practical, and easy to apply, yet still brings improvements in accuracy and explainability across different tasks. The writing is clear and the motivation is easy to follow.

**Weaknesses:**

1. Although the paper mentions the Information Bottleneck, the method does not really optimize an IB objective in practice.
2. The algorithm novelty is limited. The core idea of selecting samples based on disagreement between local and global models closely follows prior work such as FedAL and KSAS. The main difference is adding latent-space disagreement on top of prediction-level disagreement, which is a reasonable extension but still incremental, it is not introduce a new federated active learning mechanism, but just give a better score.
3. The experimental setup relies mostly on small and relatively easy datasets, and the data appear close to IID. The large-scale, highly non-IID, and diverse across clients data are not included in this paper.

**Questions:**

1. The author provide ablations for $\beta=0$ (latent-only) vs $\beta>0$ (latent+logits). Could the author also report a logits-only variant to fully isolate the contribution of each component?
2. In Figure 2a&d, the accuracy on MNIST and credit datasets quickly reaches over 99 percent for all methods, leaving little room to observe meaningful differences. What is the intended insight from these plots?

---

### Official Review · Reviewer_woxf · 2025-11-03

**Soundness:** 2
**Presentation:** 2
**Contribution:** 2
**Rating:** 2
**Confidence:** 3

**Summary:**

This paper address the problem of explainable federated active learning: "explainable" involves learning a latent representation; "federated" means the learning is done in a federated setting; "active" means that unlabeled data are selected for labeling. The paper proposes Fed-MADs,  an information-bottleneck–inspired framework for explainable federated active learning. The information-bottleneck is used to inspire a training objective and is used to do local data selection. Experiments on 4 datasets demonstrate the effectiveness of the proposed method.

**Strengths:**

The problem of explainable federated active learning is complex: it shares the challenges from federated learning, active learning and explainable AI. This paper provides a method that aims to address this problem setting. Empirical results show improvement of the proposed method over baselines.

**Weaknesses:**

Overall, there are some concerns regarding the design of the method and its evaluation.
* As the authors have acknowledged, the proposed method likely heavily relies on the quality of the global model. However, this creates a potential circular problem: the global model needs to be already good in order to find good unlabeled data to improve itself. Therefore, it is important to investigate learning performance for a wider range of number of queries. However, the experiments in the paper only cover a limited range, as illustrated in Figure 1 (a) and (d), where the accuracy has already exceeded 99.8%. Scientific research should also include investigations on scenarios where the proposed method fails, rather than only showing its successful outcomes.
* The proposed method only selects top-b scores data for label queries (e.g. line 10 in Algorithm 1). This creates concerns on whether the method is effective for large numbers of b, as it may include similar data with the same high scores. In the experiments, only b=5 is studied, which raises the question of whether the proposed method generalize to larger label query batches.

**Questions:**

In figure 3, it seems that different $\beta$, ranging from 0 to 1, leads almost identical model accuracy. Does it mean that $\beta$ is irrelevant to model accuracy?

---

### Meta-Review · Area_Chair_TkFc · 2026-01-06

**Summary:**

This paper proposes a federated active learning strategy that queries unlabeled samples where local vs. global models disagree, measured both in latent representations and predicted labels.

All reviewers agree the problem setting (label-efficient explainable FL) is relevant, and multiple reviewers found the approach simple and practical, and empirically promising on the reported benchmarks.

That said, there is also broad agreement on key weaknesses that limit confidence in the claims:

(1) the IB connection appears heuristic, with a critical approximation step that is not convincingly justified;

(2) the core mechanism resembles prior disagreement-based FAL (e.g., local/global disagreement), making the novelty feel incremental (beyond adding a latent space term and the IB context);

(3) the evaluation is too limited for the federated setting. They use largely small/near-IID datasets with very high accuracies, limited exploration of query budgets, batch sizes, and non-IID client heterogeneity. There are also missing comparisons to some more recent baselines.

Overall, the current evidence does not sufficiently back up the paper’s stronger positioning as an IB-derived, broadly applicable FAL method for realistic FL regimes.

**Reviewer Concerns:**

I did not see an author rebuttal/discussion here. So most points remain outstanding.

**Reviewer Scores:**

n/a

---

### Decision · Program_Chairs · 2026-01-26

Reject